# Triggering Organic Growth: A Fresh Challenge to Behaviour Change

**Glenn Laverack** 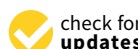

Institute of Public Health, University of the United Arab Emirates, Al Ain 17666, UAE; grlavera@hotmail.com

**Abstract:** The purpose of this paper is to discuss behaviour change beyond communication to trigger "organic growth"—a marked increase in the competencies, skills and knowledge in communities, societies and local economies. The paper discusses the challenge of triggering organic growth to help communities to build their capacity through "organic change"—concerted actions at an individual or community level to gain control over the social, economic and political influences that are necessary to improve people's lives and health. The paper discusses how organic change sometimes involves an emotional or symbolic response that can be triggered by an evidence-based argument as part of a behaviour change approach. The paper concludes that it is useful to visualise behaviour change in a fresh way that goes beyond communication to articulate capacity building and community action, and that this is best represented through the terms "organic growth" and "organic change".

**Keywords:** behaviour change; empowerment; organic growth; organic change; capacity building; policy

## 1. Introduction

The behaviour change approach promotes health through changes in lifestyle that are appropriate to people's settings, for example, to increase physical activity or to promote healthy eating. Behaviour change communication is a widely used intervention manifested through approaches such as communication for development (C4D), water, sanitation and hygiene (WASH), and social behaviour change communication. These approaches attempt to provide information and skills that people need to adopt a healthier lifestyle and use a range of techniques, including interactive technologies, motivation, counselling and persuasion. The assumption is that, before people can change their lifestyle, they must first understand basic facts about a particular health issue, adopt key attitudes, learn a set of skills and be given access to appropriate services [1]. The simple logic is that some behaviour leads to ill-health and, therefore, providing information, primarily through communication, is an effective way to reduce illness because it persuades people to change their unhealthy behaviours.

The evidence about the effectiveness of behaviour change—what works and what does not work—is unclear, although we do know that single interventions that target a specific behavioural risk have little impact on the determinants that actually cause poor health, especially for vulnerable people [2]. The popularity of government-funded single interventions, such as promoting physical activity, are attractive to decision makers because they promise quantifiable results within a short time frame, are relatively simple and offer savings in healthcare services, especially for people suffering from chronic diseases [3]. However, what is clear is that single interventions simply take our attention away from a complacent policy agenda that creates the conditions of poor health in the first place. For everyday living, this can mean minimum standards of welfare payments and reduced access to important preventive and social services [4].

What is required is more inclusive planning that—crucially—involves communities gaining more control over the determinants of their health. However, when civil society is weak, it is not always

clear whose responsibility it is to ensure that communities have every opportunity to become actively involved, for example, in the delivery of preventive and healthcare services, and to gain more control over their lives and health. Community involvement is complex, and has historically led to the use of a wide range of communication approaches based on improving participation. The purpose of this paper is to discuss a fresh way of visualising behaviour change, one that goes beyond communication, to trigger "organic growth" and to actively promote the involvement of communities.

## 2. What Is "Organic Growth"?

Organic growth is defined here as "a marked increase in the competencies, skills and knowledge in communities, societies and local economies to allow people to better organise and mobilise themselves towards achieving desired changes". Organic growth is a dynamic organisational process that builds capacity and can lead to actions to improve people's health, both at the individual and collective levels. The success of utilising "organic organisers"—local leaders amongst poor people—in the Philippines is an early example of facilitating organic growth at the community level. The local leadership style and skills included an understanding of the cultural context and was found to be an important factor influencing the way communities grow [5].

Organic growth includes an expansion of the participant base, such as the membership of civil society organisations and increased outputs, such as improved skills, styles of leadership and involvement in the delivery of local essential services. Organic growth is not isolated to a single outcome because the competencies that are developed—the skills, knowledge and organisational ability—can be used to address other issues as the circumstances and needs in people's lives change. For example, the Stadtteilmütter (Neighbourhood Mothers) project in Neukölln, Berlin, illustrates how organic growth can be facilitated by an agency in collaboration with the community. The project operated on the principle that the people who are the most capable of quickly building a meaningful relationship with migrant mothers are those who have shared similar experiences, such as other migrant mothers. The relationship helps to build trust and the confidence needed to ask questions, get answers and become receptive to change. The approach uses a peer education strategy in which the mothers first meet informally to talk about their everyday needs in regard to their children and families, as well as their education, health and wellbeing. Then, the mothers meet on a regular basis in a small group setting to discuss specific challenges, such as the health and social care services that are available in their community. The project cooperated closely with local childcare centres, healthcare services and youth centres. The sustainability and the real success of the approach lie in the way it empowers women by increasing the interaction of migrant families with local service providers [6].

## 3. What Is the Challenge of Behaviour Change and Organic Growth?

The challenge of behaviour change is to go beyond communication to trigger organic growth, such that communities can have an impact on the determinants that actually cause poor health. To do this, communities must be able to influence societal norms and behaviours, policy and legislation. Organic growth begins by building capacity towards gaining the relevant information and skills necessary in achieving specific goals through strategic planning, often for a pre-identified concern, enabling people to define, assess, analyse and then to act.

Behaviour change approaches can use a range of tools to build capacity. In particular, the "domains approach" is internationally recognised, and has been used in many settings and cultural contexts. The nine domains to build community capacity are community participation, problem assessment capacities, local leadership, organisational structures, resource mobilisation, links to other organisations and people, ability to "ask why" (critical awareness), community control over programme management and an equitable relationship with outside agents. The domains approach can be best used in public health programmes in collaboration with communities, most often with a group of representatives, normally 10–15 people, in a facilitated "workshop" setting appropriate to the cultural context [7]. The Altogether Better Project provides an example of a collaborative partnership—in this case, in the United

Kingdom—aimed to empower communities to improve their own health and wellbeing. The overall aim was to build capacity in communities and to extend the skills and expertise of local volunteers. The Project identified practical lessons: (i) The importance of securing adequate financial resources to pay for the staff involved, which proved to be very time intensive; (ii) the need to involve a wider pool of volunteers to expand the chances of success of the project; (iii) the need to define clear and more specific goals in order to make it easier to recruit volunteers for its delivery; and (iv) the need to collect data on people's personal assets in a systematic way in order for it to be usable [8].

Behaviour change approaches can, therefore, trigger organic growth by using a capacity-building approach and this, in turn, helps to mobilise people towards achieving organic change. However, it is also important that community members—especially the vulnerable, such as migrants—understand the cultural context and are able to acquire the necessary knowledge and skills to be able to engage with society. Performance art theatre can help to break down cultural barriers by helping people to see "the other" as similar to themselves, and to recognise the value of diversity as a resource and an opportunity. Compagnia Teatro dell'Argine in Italy uses participatory theatre to remove physical and psychological barriers and prejudices by including migrants and non-migrants in performances to foster inclusiveness (http://www.itcteatro.it/). Migrants are then better able to share their emotions with others, build friendships and develop a better understanding of the society in which they live.

## 4. Triggering Organic Change

Politicians are sensitive and react to the pressure applied through the actions of civil society, for example, from pressure groups and movements. Organic growth has had an important role in building the capacity of social movements to help address issues of social injustice, for example, in regard to birth control and breastfeeding [9], giving women more control over their lives and health. Women's pressure groups in the United Kingdom successfully campaigned for more funding for the use of Herceptin®to treat breast cancer because the minimum cost to pay for the treatment was well beyond the means of most patients [10]. However, organic growth, alone, is insufficient to have a broad influence without the direct actions of society, communities and local economies, and this is achieved through "organic change".

## 5. What Is Organic Change?

Organic change is defined here as "concerted actions at an individual or community level to gain control over the social, economic and political influences that are necessary to improve people's lives and health". The process involves action that is directed at achieving changes in behaviours, in the level of active participation and, most significantly, in the ability to have a social and political influence, such as on societal norms or legislation. Being motivated to take action can sometimes involve an emotional or symbolic "trigger" at an individual level, such as experiencing an illness or being involved in an accident [11]. However, people are also open to argument that is based on sound scientific evidence, and this can sometimes play an important role in a response to addressing a specific issue. Organic change is therefore achieved through motivational or politically orientated action that can be triggered by behaviour change approaches.

Individually, persuasion, skills training and developing a dialogue are useful approaches to help trigger organic change. "Healthtrainers", for example, is a free health coaching service funded by the UK National Health Service to help people to make positive and sustainable lifestyle changes in regard to weight loss, healthy eating, physical activity and alcohol consumption. The health trainers offer one-to-one advice for up to three months, on a weekly basis, to discuss progress towards achieving actions (organic change) based on personal goals and to resolve any barriers to changing behaviour, such as gaining better access to exercise facilities or to joining a support group for weight loss [12]. Group work can therefore help to facilitate organic change, for example, in women's groups for achieving improvements in the health of newborn infants, children, and mothers in South Asia and

sub-Saharan Africa [13]. By participating in groups, individuals can better define, analyse and then, through the support of others, collectively act on their shared concerns.

Community empowerment builds organically from the individual, to the group, to a wider collective as people become more concerned with addressing the social and political causes of powerlessness and poor health. This must come from within an individual, group or community, and cannot be given to them. For example, community-led quarantines have proven to be an important factor help to trace new contacts and outbreak cases. In Liberia, quarantines were most effective when led at the community level, coordinated by local people and religious leaders. There were variations in how communities implemented the quarantine, sometimes at a household level, and in other instances at the community level, but usually those quarantined were given access to water and basic provisions to help minimise violations. Raising the awareness of communities about the benefits of quarantines and giving them control (empowerment) in helping to prevent further cases was, therefore, essential to achieving success in an infectious disease outbreak [14].

If behaviour change remains at the individual or small-group level, the broader conditions that create poor health would not be resolved. Behaviour change can play an important role by going beyond communication to trigger organic growth and to move people forward to become better organised and more politically aware. This empowers people to take action and triggers organic change that can lead to a wider collective influence on health through social and political change.

## 6. Conclusions

Behaviour change uses the logic that providing information to people is an effective way to reduce illness because it will persuade them to change their unhealthy lifestyles. Whilst it is not clear what works, we do know that it is crucial to more actively involve people and communities. This is based on improving participation, and sometimes lacks clarity about who is responsible to ensure that communities have every opportunity to become involved in the delivery of services and to gain more control over their lives. It is therefore useful to articulate behaviour change in a way that goes beyond communication and to use terms that acknowledge the need to help people to empower themselves and to address the broader determinants of health. This is best represented through the terms "organic growth" and "organic change", because they capture the capacity building and community action that will lead to better health outcomes for everyone.

**Funding:** This research received no external funding.

**Conflicts of Interest:** The author declares that there is no conflict of interest.

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
