# Peer review of "Triggering Organic Growth: A Fresh Challenge to Behaviour Change"

_challenges, doi:10.3390/challe10010027_

Round 1

Reviewer 1 Report

The paper is novel in terms of content and approach to change behaviour within communities . I believe the paper can be publishes as is. Being a more intellectual theoretical paper, it would be helpful to show a figure or  diagram  that depicts both concept and their interconnectedness. If this is not possible in this paper perhps the author can present later in different articles 

Author Response

THANK YOU VERY MUCH FOR YOUR USEFUL COMMENTS. I FULLY APPRECIATE YOUR COMMENT ABOUT USING A FIGURE BUT ON REFLECTION AS THIS PAPER CONCERNS TERMINOLOGY I DO NOT THINK THAT THIS IS THE MOST APPROPRIATE PLACE TO PUBLISH. BUT I WILL CONSIDER THIS FOR OTHER PUBLICATIONS IN THE FUTURE.

Reviewer 2 Report

This commentary highlights how and why organic growth and change can influence health behaviors at the community level. I believe a wide range of case studies in various countries will help researchers and practitioners to better understand how community empowerment can be shaped and reinforced by innovative ideas and committed efforts from the community. If the author can add more contents within the word limit, I suggest the following. First, different strategies by local contexts should be delineated in detail since community organizing and its governance in participatory health programs may be varied. Second, the theoretical framework can be used to tease apart basic and/or complementary components of interventions by local contexts.

Author Response

THANK YOU VERY MUCH FOR YOUR COMMENTS.I HAVE GIVEN THEM A GOOD DEAL OF CONSIDERATION. MY INTENTION IS TO KEEP THIS COMMENTARY SHORT AND TO FOCUS ON THE CONTEMPORARY TERMINOLOGY. I AGREE THAT A LONGER PAPER WITH CASE STUDIES AND WIDER THEORETICAL INTERPRETATION COULD BE USEFUL BUT DO NOT FEEL THAT THIS IS THE RIGHT PLACE TO PUBLISH, HOWEVER, I AM CONSIDERING THIS TOPIC FOR ANOTHER FUTURE PAPER.

Reviewer 3 Report

Some parts of the document have red signed words, although I don't know the reason, if it is the doubt of include it or not, I agree to include it as they are written.

Excelente paper. Congratulations.

Author Response

THANK YOU VERY MUCH FOR YOUR CONSIDERATION. THE RED UNDERLINE REFERS TO SPELLING WHICH I HAVE DOUBLE CHECKED AND IT IS OK FOR THE ENGLISH LANGUAGE.

This manuscript is a resubmission of an earlier submission. The following is a list of the peer review reports and author responses from that submission.

Round 1

Reviewer 1 Report

This commentary advocates for “organic change” in health behavior change. I am unclear how is organic growth” different from community mobilization, a practice that has been in use for decades. Many of the concepts the author provides are similar to the environmental justice movement and other social justice movements.

Abstract

·         “The paper discusses how organic change sometimes involves an emotional or symbolic response that can be triggered by rational argument as part of a behaviour change approach.”

This statement should be revised, especially “that can be triggered by rational argument”. I am unclear what the author means by this statement.

Part 1 (Introduction)

·         “Behaviour change communication is a widely-used intervention manifested through approaches such as communication for development (C4D), water, sanitation and hygiene (WASH) and social behaviour change communication.”

I recommend using the term “health behavior change” not “behaviour change communication” to make it more applicable to the fields of public health and nursing.

I recommend also providing more specific approaches that are applicable to several fields. For example, would this include tobacco cessation? Attempts to increase physical activity or healthy eating?

·         “They use a range of techniques including interactive communication technologies, motivation, counselling, persuasion, influencing social norms and even coercion.”

“Coercion” is a very strong word and its use in behavior change is unethical. If you keep the word, I recommend providing references that shows evidence of its use, as well as references for each of the other techniques.

·         “The evidence about the effectiveness of behaviour change —what works and what does not work—is unclear.”

Provide references for this statement because I would argue that there is strong evidence that behavior change interventions that are culturally tailored to specific populations and conditions are effective.

·         I recommend using the term, “social determinants of health,” because it seems like that it was you are referring to when you discuss addressing “the conditions of poor health.” The World Health Organization has a very comprehensive definition of the term.

Part 2

·         Provide references for the nine domains to build community capacity.  

Part 3

·         Demonstrate how the “Healthtrainers” helps to trigger organic change. That is not clear from the text.

Reviewer 2 Report

I think the main premise of this paper is very important, but there are a few structural elements of the argument that don't work currently. You elide a few important philosophical questions, which I'm actually fine with, but to do that effectively, you need to be clearer about what the delineation between communication and behavior is. Communication as a technical or transactional exchange between discrete individuals is indeed the basic metaphor for much health promotion, but lots of folks have called it out. The more important question is what does it mean to change that paradigm. That certain types of non-verbal, or non-technical, or non-transactional acts could serve to trigger an organic transformation seems promising to me, but I'm not sure why it wouldn't be just another form of communication - i.e., I don't see why it's supposed to be in opposition to "communication" as such. More importantly, what it means to be an effective replacement for the traditional and impoverished sense of communication really needs to be fleshed out. The examples of holistic and capacity-building programs is too high-level to match the initial claim. Didn't they explain themselves to their target audience using some sort of communication, after all? Shouldn't you be talking about specific acts of communication, in the end, and explaining how they do or do not trigger the organic change? It might be that a diagram could help, although perhaps in that case you'd have to have overlapping Venn diagrams, with communication somehow in an overlapping space instead of a clear distinction between communication and organic growth. Or maybe communication would be the encompassing space, with merely transactional communication opposed to the communication that triggers change?